# Is YouTube promoting the exotic pet trade? Analysis of the global public perception of popular YouTube videos featuring threatened exotic animals

**Georgia Kate Moloney**[1], **Jonathan Tuke**[2], **Eleonora Dal Grande**[1], **Torben Nielsen**[1], **Anne-Lise Chaber**[1]*

1 School of Animal and Veterinary Sciences, The University of Adelaide, Adelaide, South Australia, Australia,
2 School of Mathematical Sciences, The University of Adelaide, Adelaide, South Australia, Australia

* anne-lise.chaber@adelaide.edu.au

**Data Availability Statement:** All relevant data are within the paper and its Supporting Information files.

## Abstract

The exploitation of threatened exotic species via social media challenges efforts to regulate the exotic pet trade and consequently threatens species conservation. To investigate how such content is perceived by the global community, mixed model sentiment analysis techniques were employed to explore variations in attitudes expressed through text and emoji usage in public comments associated with 346 popular YouTube® videos starring exotic wild cats or primates in 'free handling' situations. Negative interactions between wild cats and primates with other species were found to be associated with both text and emoji median sentiment reduction, however were still accompanied by a median emoji sentiment above zero. Additionally, although a negative trend in median text sentiment was observed in 2015 for primates, an otherwise consistent positive median text and emoji sentiment score through time across all IUCN Red List categories was revealed in response to both exotic wild cat and primate videos, further implying the societal normalisation and acceptance of exotic pets. These findings highlight the urgency for effective YouTube® policy changes and content management to promote public education and conservation awareness, whilst extinguishing false legitimisation and demand for the exotic pet trade.

## Introduction

Unsustainable trade in wildlife is an important challenge to species biodiversity, where live animals are trafficked for pets amongst other purposes [1]. Exotic pets are animals without an extensive history of domestication or life in captivity that are not traditionally viewed as companion animals [2]. The exotic pet trade encompasses the global exploitation of exotic animals including popular mammalian species such as primates and carnivores (including *Felidae*) commonly sourced from wild populations, thus imposing a significant threat to species biodiversity, animal welfare and public health [2–4]. Although international wildlife trade regulations are in place through the Convention on International Trade in Endangered Species of

**Funding:** The authors received no specific funding for this work.

**Competing interests:** The authors have declared that no competing interests exist.

Wild Fauna and Flora (CITES), much of the trade still operates illegally [1, 3, 5]. Therefore, modern drivers and routes of the exotic pet trade must be identified to facilitate better regulation and preserve wild populations of these species.

Consumer interaction with the exotic pet trade has drastically changed in association with widespread access to the internet, wherein social media has been identified as an important driver of the trade through significantly influencing public perception [2]. Social media has changed how society access, consume and share information, wherein exposure to content is individually tailored to enhance user experience [6]. Social media platforms have become a dominant source for news, entertainment and information, therefore understanding factors that generate appeal for these sites is important to enable content regulation [7, 8]. Currently, a considerable amount of published content remains unregulated due to poorly established policy guidelines and limited enforcement disproportionate to the volume of data uploaded [9]. Overall, social media platforms permit public accessibility to unregulated exotic animal content, influencing user perceptions and facilitating accessibility to the exotic pet trade like never before.

Investigating how social media platforms influence public perception is vital to understanding attitudes towards exotic animal content and recognise the impact these platforms have in promoting consumer demand and enabling the globalisation of the market. With more than two billion users worldwide as of 2020, YouTube® is the largest global online video website and is thus considered to be one of the most influential social networking platforms [10, 11]. YouTube® enables users to upload, create, share and engage in content featuring exotic animals, thus promoting global participation [6]. Individuals accessing unregulated videos on YouTube® commonly choose to express their opinions and approval or condemnation of content through written language and emoji usage within the associated comments section [12]. Therefore, these comments may indicate how the public perceives content depicting exotic animals in a variety of 'free handling' situations, defined as having a relationship or interaction with humans, and further suggest how this perception is altered throughout the life of a video. Sentiment analysis, being the study of opinions, attitudes and emotions conveyed via text or emoji usage, has been used to evaluate positive or negative intentions associated with viewer comments [7, 13, 14]. It has been suggested that YouTube® is used to influence public perception surrounding 'cute' exotic animals and thus promote demand for exotic pets, however the significance of this platform as a driver of the exotic pet trade has not been fully investigated [15]. Therefore, sentiment analysis techniques applied to YouTube® viewer comments could provide an indication of how such content is perceived by the public and highlight areas requiring enhanced regulation to preclude the normalisation of exotic pets.

The aims of this study were to explore public perception of exotic wild cat and primate species in 'free handling' situations featured in popular YouTube® videos and investigate variations in perception associated with time, conservation status and interactions with other species through sentiment analysis techniques. Following this, the impact on the exotic pet trade and species conservation will be discussed where YouTube® policies encompassing animal usage will be assessed accordingly.

## Materials and methods

### Online data collection

Due to their popularity in the exotic pet trade, wild cat and primate species formed the basis of the video collection [2]. Common wild cat and primate species kept as exotic pets were manually researched, predominately through Google® and with reference to the International Union for Conservation of Nature (IUCN) Red List [16], and utilised as search terms within

YouTube® (Fig 1). Popular terms which formed the basis of the video search included: 'cheetah', 'tiger', 'lion', 'leopard', 'jaguar', 'monkey', 'capuchin monkey', 'marmoset monkey', 'primate', 'chimpanzee', 'slow loris', 'spider monkey', 'macaque', 'lemur' and 'gibbon'. Further variations in search terms included the addition of seemingly popular key phrases, including 'pet', 'cute' and 'baby'. A full list of search terms investigated is located within the S1 Table. It is believed the majority of videos which feature this description were captured using these search terms.

Inclusion criteria involved the portrayal of an exotic animal in a 'free handling' setting, namely either within a captive (i.e. petting zoo) or domestic (i.e. pet) environment, interacting in some manner with either a human or another species. Videos were only considered if they had accumulated a minimum of one million views at the time of collection to ensure they had been received by a significant audience. Videos were excluded from the study if the associated comments section had been disabled. Videos published between May 2006 and October 2019 were included within the study.

Once selected, the following information was manually extracted and entered into a Microsoft Excel spreadsheet: Video web address (URL), search term used, species identified, IUCN Red List status, CITES Appendices listings, setting (captive vs pet) and species interactions present. Species were determined through analysis of the animal(s) featured in comparison to images and information sourced through online research (via Google®) and the IUCN Red List, unless specified within the video title or description. In regards to the primate videos, not all macaque and gibbon species were readily identifiable due to limitations imposed by video quality and duration. The identifiable species for both were classified as either 'Vulnerable' or 'Endangered', hence the genera were generally represented as being 'Endangered' to reflect the most severe classification.

All unnatural interactions between the exotic animal featured and other species (including humans) were classified as positive, negative or neutral. Interactions involving elements of aggression or fear were categorised as negative, while situations wherein the animal and other species were seen next to each other but not directly interacting (i.e. sleeping next to a woman talking) were considered neutral. Interactions where people were cuddling exotic animals, treating them like kids (i.e. monkey dressed up like a baby drinking a milk bottle) or otherwise promoting anthropomorphism were categorised as being positive.

Data management and analysis was conducted in R v. 3.6.1 [17]. Additional datum was extracted through the *tuber* package [18] to fulfil the following categories: Video ID, comments, date published, date downloaded, view count, like count, dislike count and comment count. Code was designed to bypass the 100 comment limitation imposed by the tuber::get_comments() function to enable all comments to be extracted. All public primary (main comments within the comment section) and secondary (comments posted in response to primary comments) comments were extracted, including those posted by the account that shared the video. The data was arranged to ensure each row represented an individual comment for analysis using the *tidytext* package [19], accompanied by the language detected through the *textcat* package [20]. Video information and comments were extracted from May 2006 to October 2019. In accordance with the aims of the study, the comments produced over time for each video were evaluated for sentiment based on language (English only) and emoji usage. 1-grams were used due to the small number of words within most comments. Tokens were neither stemmed nor lemmitised as it was considered the sentiment software would cope with the common forms of most words.

A multiple-response approach was applied to analyse species frequencies, taking into consideration the number of instances a particular species was featured across all videos within their assigned animal group. For example, when selecting for videos using primate specific

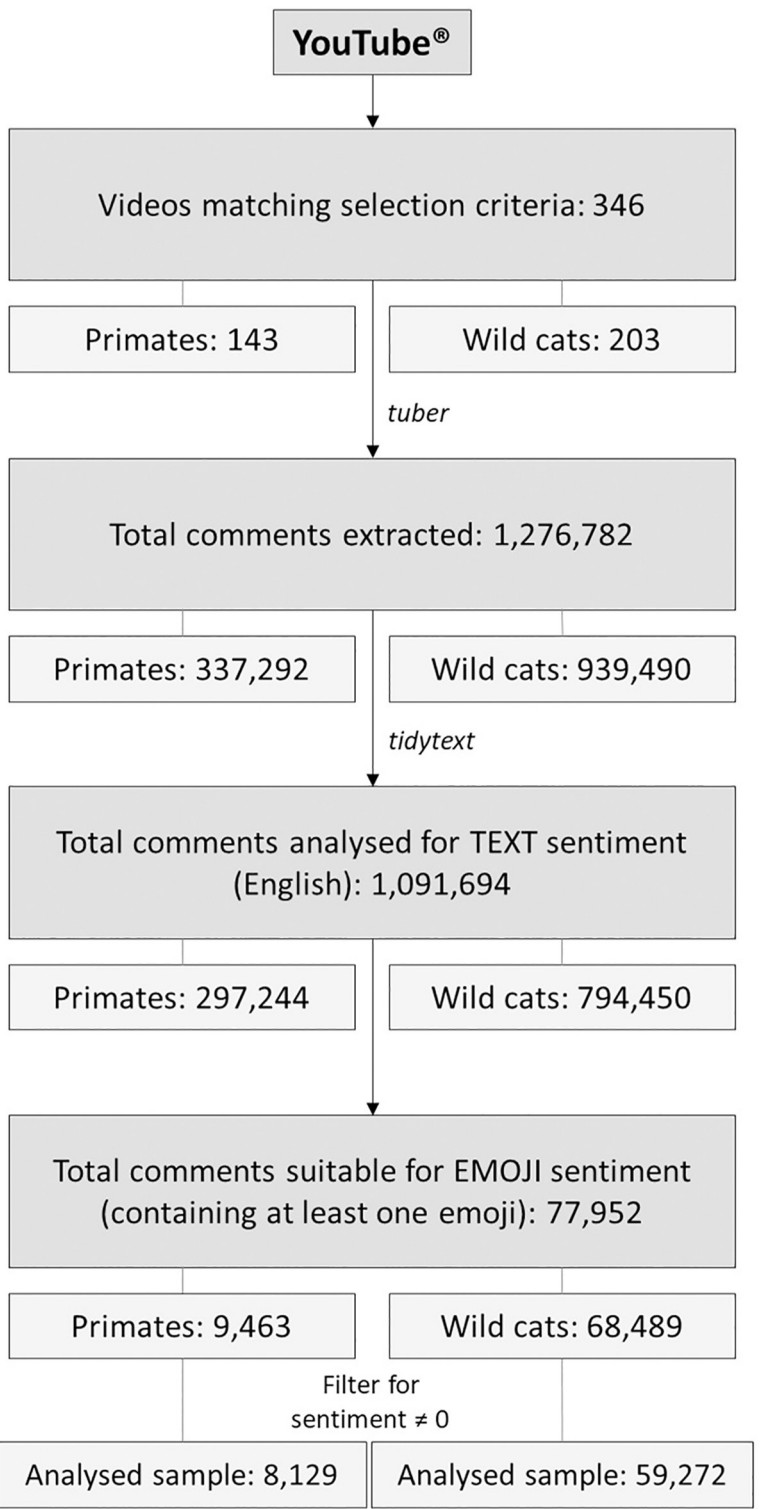

**Fig 1. Methodological framework for comment extraction and analysis.** Methodological framework for comment extraction and analysis across the 346 YouTube® videos published from 2006 to 2019, sourced in accordance with pre-determined selection criteria. Only comments extracted written in English, or containing terms identified by the sentiment software employed, were analysed for text sentiment. Likewise, only comments containing at least one emoji were analysed for emoji sentiment. Filters were applied to ensure comments analysed for emoji sentiment contained at least one emoji and had an associated positive or negative sentiment score.

search terms some videos included multiple primate species, thus each primate species appearance was listed as an observation. Twelve primate videos were considered to be 'compilation videos' as they featured four or more videos compiled together. In these instances, species and IUCN Red List classification were not specified due to ambiguity.

Sentiment analysis techniques were employed to overcome limitations imposed by traditional manual qualitative techniques and enable more efficient analysis of a large dataset [21]. Lexicons derived from the *tidytext* package were applied to impart values to key words [19, 22]. Due to software limitations, only English comments were analysed for text sentiment. The "bing" lexicon qualitatively analysed terms, denoting words as either 'negative' or 'positive', and was thus used to analyse word frequencies. The "bing" lexicon was selected as the binary categorisation enabled the most popular positive and negative sentiment terms to be identified to evaluate word frequencies and cultural trends. Similarly, the "afinn" lexicon provided negative or positive values between -1 and +1 to particular pre-determined sentiment terms to quantitatively analyse variation in median text sentiment across comments. In association with this lexicon, negative values represent negative sentiment terms, zero relates to neutral terms and positive values correspond to positive terms. The "affin" lexicon was chosen as it fulfilled the aims of the study and was the most relevant option, being based on data derived from the social media platform Twitter [23]. Individual sentiment terms were assigned a quantitative value via the "afinn" lexicon to provide an average sentiment score between -1 and +1 for each viewer comment. These average sentiment scores for individual comments were collated to derive an average text sentiment score across all wild cat or primate comments for the year in which they were published. The *ggplot2* package [24] was employed to plot variations in median text sentiment over time, in accordance with the year comments were published. Trends in median text sentiment were compared with emoji sentiment and changes in average trends over time were analysed.

Comments containing emojis were extracted and analysed within R [17]. The emoji sentiment lexicon utilised in this paper was based on work conducted by Novak et al. [14] and Peterka-Bonetta [25] as it best suited the aims of the study and provided a sentiment scale comparable to the "afinn" text lexicon used. The code provided by Novak et al. [14] was transferred directly into R and applied to the dataset without altering the sentiment data. The lexicon provided an average sentiment score for each emoji-containing comment as a value between -1 and +1, wherein negative values represent negative emoji usage, zero relates to neutral emojis and positive values correspond to positive emojis. Filters were applied to ensure comments analysed contained at least one emoji and a sentiment value not equal to zero. Changes in median sentiment score associated with emoji usage over time (based on year of comment publication) were analysed in relation to species and conservation status through the *ggplot2* package [24]. Trends in median emoji sentiment scores through time were graphically analysed in conjunction with median text sentiment scores.

Mixed-effects models were developed to assess associations between sentiment and video characteristics. The lmer function which employed, which is a general linear mixed model. Therefore the commands used to run the mixed models were 'lmer(emo_sent ~ interaction_-type + (1 | youtube_ID), data = comments_sent_clean)' and 'lmer(text_sent ~ year + interaction_type + IUCN + (1 | youtube_ID), data = comments_sent_clean)'. Sentiment for each comment, either emoji or text, were set as response variable and predictors included species, year, interaction type and IUCN red list status. A random intercept for each video was included to handle the repeated measurement of the data. Best models were selected using AIC and forward selection.

## Results

### Frequency analysis

In total, 346 videos were compiled for analysis with the majority (n = 203, 58.7%) categorised as exotic wild cat videos, based on the prominent species depicted and search terms through which they were sourced. The most popular species recorded was the tiger (*Panthera tigris*), making an appearance in 48.3% of all wild cat videos (Table 1). With regards to the wild cat videos, tigers attracted the greatest number of comments, followed by lions (*Panthera leo*) then cheetahs (*Acinonyx jubatus*), closely reflecting species frequencies within the dataset (Table 1). Conversely, with regards to the primate videos, vervet monkeys (*Chlorocebus pygerythrus*) attracted the highest number of comments, followed by capuchin monkeys (*Cebus apella*) and macaques (*Macaca spp.*). The number of associated comments varied per video, as specified within the S2 Table. An interesting observation was the finding that 18 of the videos had one viewer that posted over 10% of the comments to the video with one viewer posting >2,500

**Table 1. Species representation and associated conservation status.** The number of instances and percentage of corresponding YouTube® videos published between 2006 and 2019 associated with search terms explored, wherein key exotic wild cat (a, n = 203) and primate (b, n = 143) species were identified. A multiple response analysis was considered as some videos featured multiple species. The frequency describes the number of videos in which each species appeared, whereas the percentage represents the number of videos each species appeared in as a percentage of all wild cat or primate videos. Instances where primate or exotic wild cat videos featured more than one associated species have been included. Species with a frequency equal to or less than one have been excluded from the table.

| Species | Frequency | Percentage (%)[1] | CITES[2] Appendix | IUCN[3] Red List Status |
|---|---|---|---|---|
| **Wild cats[4] (a)** | | | | |
| Tiger (*Panthera tigris*) | 98 | 48.3 | I | Endangered |
| Lion (*Panthera leo*) | 64 | 31.5 | II | Vulnerable |
| Cheetah (*Acinonyx jubatus*) | 53 | 26.1 | I | Vulnerable |
| Jaguar (*Panthera onca*) | 13 | 6.4 | I | Near Threatened |
| Leopard (*Panthera pardus*) | 13 | 6.4 | I | Vulnerable |
| Caracal (*Caracal caracal*) | 3 | 1.5 | I | Least Concern |
| **Primates[5] (b)** | | | | |
| Capuchin monkey (*Cebus apella*) | 42 | 29.4 | II | Least Concern |
| Macaque (*Macaca spp.*) | 28 | 19.6 | I | Endangered[6] |
| Vervet monkey (*Chlorocebus pygerythrus*) | 8 | 5.6 | II | Least Concern |
| Chimpanzee (*Pan troglodytes*) | 7 | 4.9 | I | Endangered |
| Gibbon (*Nomascus spp.*) | 6 | 4.2 | I | Critically Endangered[6] |
| Pygmy slow loris (*Nycticebus pygmaeus*) | 7 | 4.9 | I | Vulnerable |
| Orangutan (*Pongo spp.*) | 6 | 4.2 | I | Critically Endangered |
| White faced capuchin monkey (*Cebus albifrons*) | 6 | 4.2 | II | Least Concern |
| Green monkey (*Chlorocebus sabaeus*) | 5 | 3.5 | II | Least Concern |
| Spider monkey (*Ateles spp.*) | 5 | 3.5 | I | Endangered |
| Pygmy marmoset (*Callithrix pygmaea*) | 4 | 2.8 | II | Least Concern |
| Gorilla (*Gorilla spp.*) | 3 | 2.1 | I | Critically Endangered |
| Black and white ruffed lemur (*Varecia variegata*) | 3 | 2.1 | I | Critically Endangered |
| Squirrel monkey (*Saimiri spp.*) | 2 | 1.4 | I | Vulnerable |

[1]Percentage of wild cat (a, n = 203) or primate (b, n = 131, excluding 'compilation' videos) videos which featured a particular species

[2]Convention on International Trade in Endangered Species of Wild Fauna and Flora (CITES) Appendix entries (2019)

[3]International Union for Conservation of Nature (IUCN, 2019)

[4]Other wild cat species featured include serval (*Leptailurus serval*), puma (*Puma concolor*)

[5]Other primate species featured include baboon (*Papio spp.*), tarsier (*Tarsius spp.*), owl monkey (*Aotus spp.*)

[6]Not all videos with *Macaca spp.* and *Nomascus spp.* contained identifiable species. IUCN listing was based on the most severe status of identified species.

comments (data not shown). The highest proportion of comments from one viewer was 46.5% (33/71). These frequent commenters can obviously influence the sentiment of the individual video. However, as they are only present in relatively few videos, they are unlikely to have affected the overall findings in this study.

Human interactions with wild cats and primates were observed in 90.5% of all videos. The most common non-human species interaction depicted was with domestic dogs, as featured in 17.9% of all videos. The predominant language detected within comments was English (49.4%). Of the comments from which emojis were extracted, emoji usage varied between the animal groups. Overall, 8.6% of wild cat comments contained at least one emoji, compared with only 3.2% of primate comments. Species IUCN Red List conservation statuses ranged from 'Least Concern' to 'Critically Endangered'. Species listed in CITES Appendices I and II were represented within the dataset, indicating imposed trade restrictions [5].

## Sentiment analysis

The most popular sentiment terms identified in both wild cat and primate comments were 'cute', 'like' and 'love'. 'Like' presented as the most frequently used term associated with wild cats, comprising 8.6% (n = 81,102) of all sentiment terms recognised, compared to 'cute' constituting 11.5% (n = 33,203) of all sentiment terms identified within primate comments. Videos of most species demonstrated a text sentiment above one (Fig 2). Primate videos were associated with the six highest median sentiment scores, closely followed by leopards (*Panthera pardus*) which reached the highest score for wild cats. In contrast to most primates, chimpanzees (*Pan troglodytes*) had a median text sentiment score close to zero and slow loris (*Nycticebus pygmaeus*) videos had a median sentiment score below zero.

Text and emoji sentiment varied little between different IUCN Red List categories for exotic wild cats and remained positive for all categories despite median emoji sentiment being lower than median text sentiment. 'Least Concern' IUCN listed primates had the highest sentiment of all categories (Fig 3), while IUCN listed 'Vulnerable' primates had the lowest sentiment and was the only IUCN group approaching neutral. This was also the only category with a neutral text sentiment and where emoji sentiment was higher than text sentiment. Videos showing

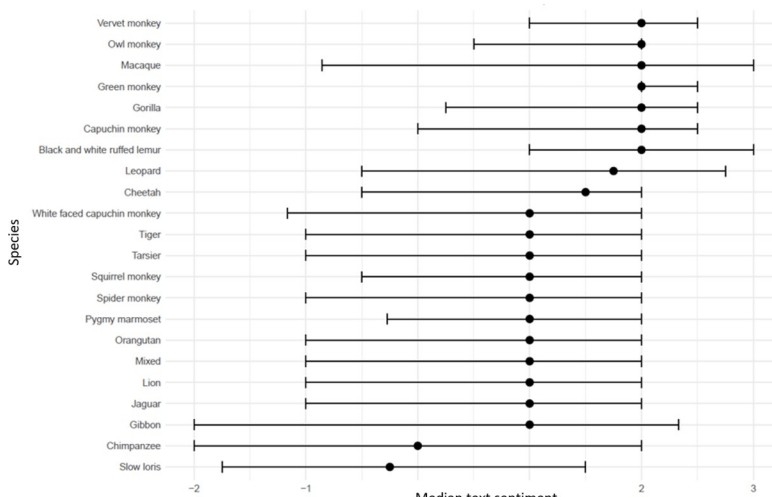

**Fig 2. Median text sentiment of 346 YouTube® videos published between 2006 and 2019 associated with search terms explored, wherein key exotic wild cat (a, n = 203) and primate (b, n = 143) species were identified.** Error bars represent first to third quartiles.

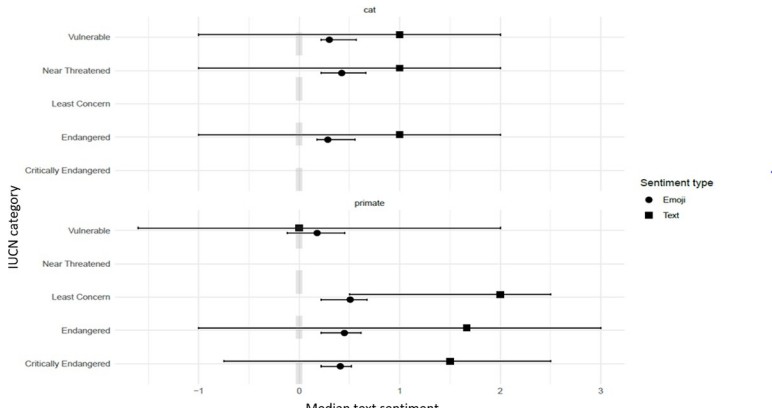

**Fig 3. Sentiment and IUCN Red List classification.** Median text and emoji sentiment comments score across 346 YouTube® videos of exotic wild cat published between 2012 and 2019 associated with species conservation status as listed on the IUCN Red List (2019). Emoji usage within the comments was only present from 2012. Twelve primate compilation videos featuring four or more compiled videos with various species were not provided with an IUCN Red List Classification due to ambiguity and as such have not been included. Median sentiment score ranged between -1 (negative) and +1 (positive), where 0 is neutral, in accordance with the software utilised. Error bars represent first to third quartiles.

positive interactions between exotic wild cats, primates and other species had a median text sentiment above one and negative interaction videos had a sentiment below zero (Fig 4). All types of interactions had median emoji sentiment above zero with negative and neutral interactions having similar median sentiment, although both were lower than for videos with positive interactions.

Median sentiment scores varied between exotic wild cats and primates but they remained above zero through time for both groups, indicating a positive sentiment (Fig 5). For cats, sentiment decreased from 2006 until 2011 where it remained stable until 2019. Primate text sentiment varied until 2011 and decreased from 2010 to 2016 after which it increased to 2006 levels.

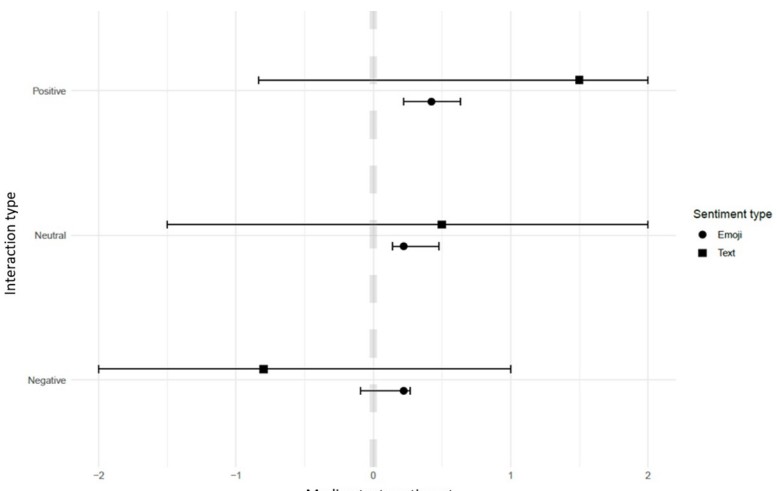

**Fig 4. Median text and emoji sentiment of 346 exotic wild cats and primates YouTube® videos published between 2006 and 2019 based on interaction between wild cats or primates with other species.** Error bars represent first to third quartiles.

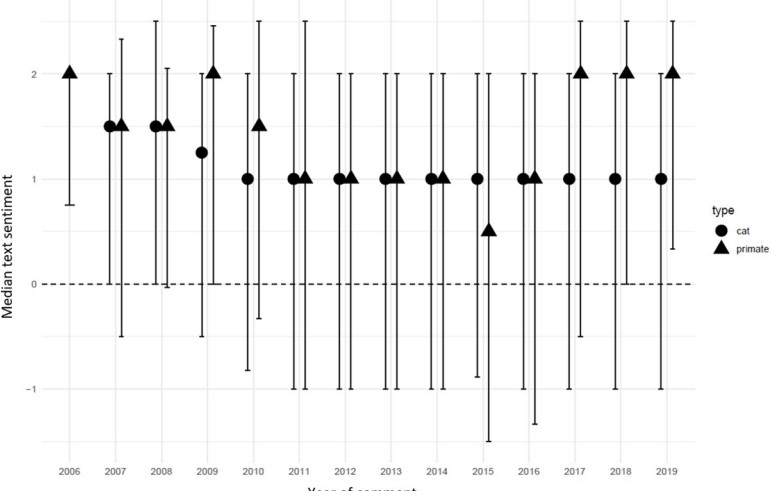

**Fig 5. Median text and emoji sentiment over time of 346 exotic wild cats and primates YouTube® videos published between 2006 and 2019.** Error bars represent first to third quartiles.

There was a decrease in primate sentiment in 2015. Emoji sentiment appeared less varied but remained positive in the available period (2012 onwards, data not shown).

Modelling of the text sentiment showed that average sentiment was 0.6 to 0.7 lower in 2011 to 2019 compared to 2006. Both positive and neutral interaction videos had 1.2 higher average sentiment than videos with negative interactions. Videos with species of IUCN red list category of 'Least Concern' had an average sentiment higher than videos with 'Vulnerable' to 'Critically Endangered' species (Table 2).

An association between interaction and average emoji sentiment was found with videos including negative interactions having an average sentiment of 0.2 less than videos with neutral or positive interactions (Table 3). Inclusion of any additional variables were not found to improve the model.

## Discussion

This study explored viewer attitudes towards exotic species in free handling situations featured in popular YouTube® videos as an indication of public perception and how variation in perception developed in different contexts. Free handling has been defined as exotic animals featuring an unnatural relationship with humans, where direct human interaction was evident in over 90% of videos analysed whilst over 17% of videos also depicted interactions with domesticated dogs. These interactions could be normalising keeping exotic pets in captivity and increase demand for these species in the pet trade [26]. In this study, YouTube® comments serve as a gauge of public perception due to the platforms wide-reaching accessibility and popularity, where several studies have supported the impact social media platforms have on public opinion [27, 28].

The prevailing frequency of positive terms, especially 'cute', 'like' and 'love', within video comments in this study reflects the use of the 'cute' factor in the promotion of online content and as a universal marketing tool for exotic pets. 'Cute' animal videos have been an integral part of YouTube® since its inception in 2005, where even the first ever video posted featured exotic animals [11]. In accordance with prior research, the 'cute' factor enhances desirability and the likelihood of consumers acquiring an exotic pet [4, 29], similar to strategies used in the sale of companion animals or even commercial products, contributing to the normalisation of

**Table 2. Variables associated with text sentiment of 346 YouTube® videos of exotic cats and primates uploaded between 2006 and 2019.**

| Variable | | Estimate | Std. error | p-value |
|---|---|---|---|---|
| Intercept | | 0.337 | 0.255 | |
| Year | 2006 | Reference | - | - |
| | 2007 | -.0179 | 0.184 | 0.187 |
| | 2008 | -0.062 | 0.166 | 0.331 |
| | 2009 | -0.218 | 0.164 | 0.709 |
| | 2010 | -0.466 | 0.163 | 0.183 |
| | 2011 | -0.681 | 0.163 | 0.004 |
| | 2012 | -0.699 | 0.163 | <0.001 |
| | 2013 | -0.708 | 0.163 | <0.001 |
| | 2014 | -0.648 | 0.163 | <0.001 |
| | 2015 | -0.664 | 0.163 | <0.001 |
| | 2016 | -0.635 | 0.163 | <0.001 |
| | 2017 | -0.707 | 0.163 | <0.001 |
| | 2018 | -0.668 | 0.163 | <0.001 |
| | 2019 | -0.769 | 0.163 | <0.001 |
| Interaction type | Negative | Reference | - | - |
| | Neutral | 1.177 | 0.159 | <0.001 |
| | Positive | 1.198 | 0.156 | <0.001 |
| IUCN listing | Least Concern | Reference | - | - |
| | Vulnerable | -0.482 | 0.098 | <0.001 |
| | Near threatened | -0.589 | 0.207 | 0.027 |
| | Endangered | -0.585 | 0.090 | <0.001 |
| | Critically Endangered | -0.431 | 0.157 | 0.006 |
| Random effect | | Variance | Std. Dev | |
| | YouTube® video | 0.328 | 0.573 | |

Only English comments are included.

exotic pets [30]. However, the popularity of the term 'like' in this context has a dual purpose as it echoes the participatory culture fostered by YouTube®, wherein users are encouraged to 'like, share and subscribe' to content [31, 32]. As 1-gram tokens were employed, the lexicon used could not distinguish between the positive sentiment term and the alternative application of the word, therefore this may provide inaccuracies in the interpretation of user sentiment. Further limitations associated with the sentiment software were identified, including the inability to assess non-English words or to recognise word usage in alternate contexts, such as

**Table 3. Variables associated with emoji sentiment of 346 YouTube® videos of exotic cats and primates uploaded between 2006 and 2019.**

| Variable | | Estimate | Std. error | p-value |
|---|---|---|---|---|
| Intercept | | 0.145 | 0.028 | |
| Interaction type | Negative | Ref | | |
| | Neutral | 0.200 | 0.031 | <0.001 |
| | Positive | 0.220 | 0.029 | <0.001 |
| Random effect | | Variance | Std. Dev | |
| | YouTube® video | 0.010 | 0.101 | |

Emojis were not available until 2012.

sarcasm [14, 33]. To overcome these limitations emoji usage was assessed, wherein the over-whelming majority of analysed comments (87.1% wild cats, 89.8% primates) supported posi-tive global public perception. Emojis effectively represent a global language of non-verbal cues used to convey the true meaning and emotion of text on social media, mirroring traditional behavioural indicators which would typically be present in face-to-face communication. For example, popular 'tongue', 'wink' and 'smirk' emojis commonly assist in the universal expres-sion and recognition of sarcasm and therefore amend inaccuracies in text sentiment analysis by acting as metadata [34–36]. Likewise, certain emojis can signify that the user identifies con-tent as being 'cute', or to express affection or happiness [37]. Both text and emoji usage dis-played a predominately positive viewer response to content as they remained above zero irrespective of species and time, suggesting positive attitudes towards these 'cute' exotic ani-mals in free handling situations.

The interaction between exotic wild cats and primates with other species were associated with both text and emoji sentiment in the mixed model (Tables 2 and 3). Negative interactions videos had lower sentiment than videos including neutral or positive interactions, suggesting that viewers are concerned about the welfare of the animals, although it cannot be ruled out that it is based on concern for the human or other interacting species in the video. An example supporting the notion that reduced sentiment is due to concern for the exotic animal is the "Tickling is TORTURE" campaign launched in 2015 aimed to combat slow loris exploitation, which had an evident impact on the perception of exotic animals featured online [38]. The suc-cess of this strategy may even be reflected in the negative text sentiment trend associated with slow loris videos (Fig 2) and primate videos overall in 2015 (Fig 5), highlighting the positive impact of public education. However, following this decline sentiment scores began to increase from 2016, possibly suggesting the awareness encouraged by the campaign was either forgotten or not extrapolated to other exotic species. Fluctuating internet trends and 'viral videos' need to be closely monitored as they challenge the implementation of strategies to successfully oppose featured content [39]. However, these finding suggests that education of the public on welfare of exotic animals in captivity could be used to influence people's perception of them as pets.

Videos with 'Least Concern' species demonstrated a higher median text sentiment than vid-eos including species in the categories 'Vulnerable' to 'Critically Endangered' (Table 2), how-ever IUCN listing was not found to be associated with emoji sentiment (Table 3). It should be noted that exchanging IUCN listing with species created very similar results (data not shown). However, the sentiment scores remained consistently positive across all species, despite IUCN Red List classification (Fig 5). Interestingly, the most viewed video included in the study dis-played 'Endangered' macaques performing for entertainment purposes (S3 Table). Despite their conservation status and evident animal welfare implications, the associated volume of views and overall positive sentiment score suggested this content was positively received by viewers. This evident lack of understanding mirrors findings presented by Stazaker and Mac-kinnon [40], where public ignorance of species conservation status may translate into creating demand for these animals. It may be argued that portraying exotic animals in free handling sit-uations, as seen in more than 90% of all videos, alludes to the supposed domestication of such animals and consequently endorses exotic pets [26, 41]. The concept of free handling blurs the line between privately and institutionally owned exotic animals as they all feature a relation-ship with humans in some capacity, thus they are treated in the same manner for the purposes of this paper. Free handling of exotic animals further negates the risk of important zoonoses such as rabies and therefore presents a significant public health risk [42]. Overall, emoji usage analysed within the study demonstrates viewer's inability to recognise and acknowledge

biodiversity and public health consequences associated with the portrayal of exotic animals in free handling situations, heightening the urgency for widespread public education.

The exploitation of wild animal populations for the exotic pet trade has significantly contributed to welfare issues in addition to loss of biodiversity [2, 29]. Due to the consistent positive public perception highlighted across all videos, it is suggested that the public are inadequately educated about the origins of featured animals and associated welfare consequences. This implies that previous programs established by conservation organisations to discourage the online exploitation of wild animals have been largely ineffective. However, in the context of the study it was not possible to accurately assess the welfare status of animals featured in all selected videos accurately as they were typically too short, of poor quality and/or did not adequately display the animal's living conditions. Education is vital to ensure viewers can identify the inappropriate depiction of exotic animals and are encouraged to report such content under YouTube®'s current guidelines [11]. The positive reception of online content featuring exotic animals interacting closely with humans further enhances demand for the exotic pet trade, therefore effective measures must be taken to appropriately inform viewers about the consequences of their support of this content.

Accessibility to the sale of exotic animals has been revolutionised through the internet, where exotic animal content posted on social media has supported the normalisation, desirability and acceptability of exotic species as pets [30]. Compared to primates, wild cats were the most popular group represented within the dataset. Tigers appeared in 48.3% of all wild cat videos in a wide range of domestic and captive situations despite their 'Endangered' conservation status [16, 43], with their popularity on social media supported by Spee et al. [26]. Exotic cats are heavily traded due to their popularity as pets, allure in captivity and demand for their parts, where the internet is involved in facilitating the trade [44]. Although majority of the species featured in this study were listed under CITES Appendix I, wherein trade is only permitted under exceptional circumstances, many have been reported in the live trade [5]. Harrington [45] described popular wild cat and primate species traded and reported via CITES including lions, tigers, leopards, jaguars, servals, squirrel monkeys, marmosets, capuchins and tamarins, many of which were sourced from wild populations. These particular species were also featured in selected YouTube® videos (Table 1), further supporting the link between social media depiction and the exotic pet trade. Trade routes for the importation and export of live animals were identified to include South America, North America, Africa, Asia and Europe, demonstrating the vast reach of the live exotic animal trade, wherein this distribution was reflected in the video origins included within this study (S3 Table) [45]. Regardless of CITES protection status and trade restrictions, there is still an extensive illegal trade present to supply the significant demand for exotic pets, much of which remains unmonitored [29]. Understanding how social media influences accessibility, public perception and consumer demand is essential to enable implementation of effective strategies to monitor the modern trade and exploitation of threatened species.

Content posted on YouTube® is loosely regulated despite the provision of policies outlining expectations and limitations, wherein the site heavily relies on consumers to report ('flag') policy violations or illegal content [11]. In the case of animal abuse, reporting under the "violent or graphic content policies" guidelines (S1 Fig) is heavily reliant on an individuals' ability to detect signs of distress or mistreatment, which may vary. As indicated by this research, the public cannot be relied upon to accurately identify the inappropriate use of exotic animals as much of the content featured was seemingly considered 'acceptable' by viewers despite underlying welfare and conservation concerns. The inaccurate interpretation of welfare is further aided by the short duration, poor quality and obscured setting of many videos, as previously discussed. YouTube®'s proprietary algorithm is designed to enhance user engagement by

directing users to suggested videos based on their previous viewership, therefore implying that once a user has shown an interest in exotic animal videos they are more likely to be directed towards similar content. The website has been progressively redesigned to focus less on advertising the most popular videos to tailoring suggested content to the individual, however corporations can also bias recommended content [40]. As YouTube® itself plays a crucial role in enabling and encouraging public access to this content, they must also accept social responsibility for creating a culture wherein human engagement with threatened exotic wildlife has become acceptable.

Development or adaptation of artificial intelligence systems which can accurately identify threatened exotic species depicted in public social media content, such as the 'Wildbook' software [46], should be explored. This technology could be linked to an automated notification which appears with information pertaining to that particular species' conservation status and the exotic animal trade before enabling the video to be viewed. The purpose of this system is to identify popular exotic animal usage for research purposes whilst providing credible information to the public, enabling viewers to make a more informed decision before accessing content or supporting the exotic pet trade, similar to Instagram®'s 'Wildlife Alert System' designed to combat the use of exotic animal photo props [47]. Alternatively, it is recommended YouTube® employ software to automatically detect key terms such as species names within video titles or descriptions and flag them for immediate review. In association with revised YouTube® policy guidelines prohibiting videos displaying interactions between humans and exotic wildlife, reviewed videos in violation of newly established policies should be promptly removed to discourage normalisation of these exotic animals as pets. Alternatively, the development of an advertisement campaign presenting information surrounding exotic animal exploitation through the pet trade and the importance of species conservation could be automatically applied before all exotic animal videos to discourage support from viewers. However, in relation to Moorhouse et al. [48], it may be beneficial to focus information around the potential zoonotic risks and legal consequences of exotic pet ownership in addition to the conservation implications to better engage viewers. Effective content monitoring can not only assist with gauging public perception to address the evident education gaps, but also help to identify new trends within the exotic pet industry and thus infer conservation and public health risks.

## Conclusions

Analysis of text and emoji usage within comments revealed a predominantly positive global public perception in response to the exploitation of exotic wild cats and primates on YouTube®. Although text sentiment was higher for 'Least Concern' IUCN Red List species, the overall positive response generated irrespective of conservation status appeared to indicate a lack of public awareness. In response, implementation of targeted YouTube® policies, provision of information for users on animal welfare issues and conservation status as well as accurate content regulation is highly encouraged to cease the illusion of the normalisation of threatened exotic animals as pets and prevent false legitimisation of the exotic pet trade.

## Supporting information

**S1 Table. YouTube search terms.** Search terms investigated through the YouTube® search engine to source videos for analysis. Only search terms which produced new videos selected for analysis are listed.
(TIF)

**S2 Table. Number of comments across all wild cat and primate videos.** The number and distribution of comments extracted between May 2006 and October 2019 across all wild cat and primate videos included within the study. The number of comments associated with each video and each identifiable species varied.
(TIF)

**S3 Table. Popular videos within the study.** The top ten most popular videos included within the study, according to view count at the time of data extraction, listed with primary species featured and additional video specifications.
(PDF)

**S1 Fig. YouTube® policy guidelines.** Excerpts from YouTube®'s "violent or graphic content" policy guidelines available online. The policy sections featured are those pertaining to the use of animals within videos.
(TIF)

**S1 Appendix. Exotic wild cat video URLs (n = 203).**
(PDF)

## Author Contributions

**Conceptualization:** Anne-Lise Chaber.

**Data curation:** Georgia Kate Moloney, Jonathan Tuke, Eleonora Dal Grande.

**Formal analysis:** Georgia Kate Moloney, Eleonora Dal Grande.

**Investigation:** Georgia Kate Moloney, Jonathan Tuke, Eleonora Dal Grande.

**Methodology:** Georgia Kate Moloney, Jonathan Tuke, Eleonora Dal Grande.

**Project administration:** Anne-Lise Chaber.

**Software:** Jonathan Tuke.

**Supervision:** Torben Nielsen, Anne-Lise Chaber.

**Writing – original draft:** Georgia Kate Moloney.

**Writing – review & editing:** Torben Nielsen, Anne-Lise Chaber.

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
