## [Decision Letter · Decision Letter 0]

7 Sep 2020

PONE-D-20-17651

Is YouTube® promoting the exotic pet trade?

Analysis of the global public perception of popular YouTube® videos featuring threatened exotic animals.

PLOS ONE

Dear Dr. Chaber,

Thank you for submitting your manuscript to PLOS ONE. After careful consideration, we feel that it has merit but does not fully meet PLOS ONE’s publication criteria as it currently stands. Therefore, we invite you to submit a revised version of the manuscript that addresses the points raised during the review process.

We look forward to receiving your revised manuscript.

Kind regards,

Jorge Ramón López-Olvera

Academic Editor

PLOS ONE

Journal Requirements:

Additional Editor Comments (if provided):

Dear Dr. Chaber,

I have now received the comments from three independent reviewers regarding your submission. While all three found your study interesting and worth publishing, they also found all that the manuscript needed major deep modifications before being considered fully suitable to be published in PLoS ONE.

I recommend you to address all the comments by the reviewers in order to gain acceptance of your manuscript.

Best luck when carrying out your review.

Reviewers' comments:

Reviewer's Responses to Questions

**Comments to the Author**

1. Is the manuscript technically sound, and do the data support the conclusions?

Reviewer #1: Partly

Reviewer #2: Yes

Reviewer #3: No

2. Has the statistical analysis been performed appropriately and rigorously? 

Reviewer #1: Yes

Reviewer #2: Yes

Reviewer #3: No

3. Have the authors made all data underlying the findings in their manuscript fully available?

Reviewer #1: Yes

Reviewer #2: No

Reviewer #3: No

4. Is the manuscript presented in an intelligible fashion and written in standard English?

Reviewer #1: No

Reviewer #2: Yes

Reviewer #3: Yes

5. Review Comments to the Author

Reviewer #1: Overall this is a useful and interesting paper that with some revision should be published. At the moment, the results read as a random list. This is perhaps because overall the literature review is a bit scanty and the literature that was reviewed was not fully developed to create research questions or hypotheses. Thus the results, although using some novel analyses, feel like a descriptive list rather than answering a series of research questions contextualised from the published literature. Once research questions are presented, then the rest should follow. Some small comments.

1. it does not seem in the contents the authors analysed if people wanted the animals as a pet - with this content analysis missing, it thus not so relevant to have a discussion about the pet trade - just because people think the material is cute does not necessarily mean they want one.

2. on that same note, a lot of the animals are probably in a very poor welfare state - this does not seem to have been recorded, but this could be a more serious discussion point since if people are looking at poorly kept sickly animals, this may be more reflected by the comments than the conservation status (see

Clarke, T. A., Reuter, K. E., LaFleur, M., & Schaefer, M. S. (2019). A viral video and pet lemurs on Twitter. PloS one, 14(1), e0208577.

Nekaris, K. A. I., Musing, L., Vazquez, A. G., & Donati, G. (2015). Is tickling torture? Assessing welfare towards slow lorises (Nycticebus spp.) within Web 2.0 videos. Folia Primatologica, 86(6), 534-551.

3. the authors need to be more clear that emojis were not really available on mobile phones until the period stated (about 2012, but actually even later than that on most phones) - so their predictions and analysis should take this into account.

The IUCN Red List status is interesting; most macaques are NOT Endangered and the ones kept as pets are likely to have been (until about last week when the new listings were published) Least Concern. It would be useful if the species could be identified.

Some random (e.g. in the absence of research questions) results come up that are interesting - like human interaction - the authors could make a better case for why this is important and link it better to zoonosis at the end (c.f. Moorhouse, which they cite)

Grammar check needed - which vs that is used incorrectly; the text is largely passive; the term data should be plural -etc.

Reviewer #2: Overall, I think this is a really good paper, clearly written, and an interesting piece of work. Well done. I have a few minor comments that I think you need to address and a general point you may wish to consider. I would recommend this paper for publication with these minor comments addressed.

Minor comments:

Within the methodology:

We need to know your text mining process in more detail, including how you tokenised the text (did you go so far as to explore n-grams before decided on tokens?)

Were your tokens stemmed or lemmatised? Either is fine, but the reader needs to know. (My personal preference is always lemmatisation).

Do you have an estimate of the penetration of your search terms, e.g. was there more out there? I think a general statement of “We feel confident we captured the majority of the videos that feature this description” / “We are sure there could be many more out there that the search algorithms could not identify” is fine.

Within the results:

This pairs with the n-gram comment in above. Your results that ‘like’ is common in the comments, could this be the repetition of ‘like and subscribe’ (the Youtube mantra?). If you only observed 1-gram tokens, you need to discuss this and highlight that ‘like’ here serves a dual purpose and may not be indicative of sentiment.

My general point:

I felt the piece was missing a bit of YouTube culture discussion, particularly as you’ve focussed on high-view count videos. The interaction between the Algorithm, the engagement on the video (the ‘like and subscribe’ problem), and how influencers engage with the material is really interesting. This is all a complicated and perhaps more cultural than you are usually familiar with, but some discussion around this would really be the cherry on the icing for me.

I wasn’t able to spot your data and code files?

I believe that signing reviews leads to a more open and transparent review process. – Jill R D MacKay

Reviewer #3: I've written up my review in markdown format. So that it is appropriately formatted and not filled with markdown syntax I have instead opted to attach the review in html and pdf format to the authors. I hope that they find them useful.

6. PLOS authors have the option to publish the peer review history of their article (what does this mean?). If published, this will include your full peer review and any attached files.

Reviewer #1: No

Reviewer #2: **Yes: **Jill R D MacKay

Reviewer #3: **Yes: **Mason Fidino

---

## [Author Response · Author response to Decision Letter 0]

9 Nov 2020

Reviewer 1

Thank you for your comments and editing suggestions.

This research project was designed as a descriptive observational study and as such no

hypothesis or research questions were developed. We wanted to explore whether there was a

relationship between the variables, however as there is limited published research surrounding

how the public respond to captive wildlife featured in online videos it seemed unreasonable to

formulate a hypothesis as we did not know what to expect. Due to the gap in the literature, the

paper was predominately a descriptive analysis. Further studies have been planned where data

will be collected based on hypotheses.

1. In accordance with research conducted by Nekaris et al. add others, the ‘cute factor’ is

commonly used as a marketing tool for pets, both domestic and exotic, as it increases the

likelihood that a consumer will acquire a pet. The link between the ‘cute’ factor and the desire

for an animal has been well established, therefore we evaluated the use of ‘cute’ and additional

positive sentiment terms to suggest a higher likelihood that these people wanted the animals

shown as a pet, not necessarily to say that they would be able to attain one. These positive

terms were also associated with the keywords used to select videos. The exotic pet trade is

fuelled by such desire and therefore the authors considered it appropriate to suggest a link

between the depiction of ‘cute’ animals on YouTube® with increased demand for exotic pets.

We have included a more thorough description of this point in line 279-281. Although it is a

valid point, full analysis of the content itself, as suggested, was not part of the study and could

not be accurately assessed based on our data.

2. The assessment of animal welfare was initially considered in the early stages of the research.

However, upon analysing videos for the study, it was not possible to accurately assess the

welfare state consistently across all videos as they were generally too short and the content

disallowed the conditions to be assessed properly as certain aspects were not shown on screen.

Therefore, it was difficult to accurately comment scientifically on welfare status, as mentioned

in line 327-330.

3. Thank you for your comment. A statement has been added to the paper to enhance clarity

regarding emoji availability. In regards to the macaque conservation status, not all macaque

species were readily identifiable due to limitations imposed by video quality and duration. The

identifiable macaques were classified as either Vulnerable or Endangered on the IUCN Red List,

hence macaques in the study were generally represented as being 'Endangered' to reflect the

most severe classification, as explained within the methods (line 101-104).

4. In regards to the seemingly random results regarding human interaction, the authors have

edited the discussion to clarify why this was important. Human interaction, either directly seen

within the video or implied as the animals were captive, was an important component of the

study as we aimed to analyse videos showing animals involved in ‘free handling’ situations (see

the discussion for further information on this, line 311-315) and to stress the importance of

zoonoses (where the discussion was edited to highlight this, as suggested). Thank you for your

suggestion to clarify this point and to link the discussion back to the zoonosis risks, this has

been considered and hopefully explained more clearly in the revised discussion. 

In regards to the grammar check, we thank you for bringing this to our attention. The paper was

edited carefully to try to overcome this.

Reviewer 2

Thank you for your time and comments.

In terms of how we tokenised the text, we decided to use 1-grams due to the smaller number of

words in most comments. This is now included in line 116-117.

We did not stem or lemmatise tokens, again due to the small number of words in each comment,

however we agree that lemmatisation is a good idea for future investigations. We considered that

the use of the sentiment lexicons would cope with the common forms of the words, as stated in line

117-118.

In regards to the penetration of search terms, we feel confident we captured the majority of the

videos that feature this description. We have added a statement within the paper (line 88-89) as

suggested.

Thank you for your insightful suggestion regarding the reflection of the dual usage of ‘like’. 1-gram

tokens were used and as such we found this comment helpful and have added some discussion

around YouTube culture to the paper (see line 281-286). We are hoping this provides the missing link

between the YouTube algorithm and user engagement.

Data and code files should now be uploaded with the revised paper as a single pdf.

Reviewer 3

1. Thank you for your insight into the social science aspect of the use of ‘perception’ vs ‘opinion’. In

this instance, we believe that the use of either term is effectively equivalent for the purposes of this

study.

2. We agree that further statistical analysis would be helpful in interpreting the results, however as

this was designed as a descriptive observational study we have decided not to go into this detail as it

was beyond the scope of the study. As such, the data collected was not associated with a hypothesis

but was collected to evaluate whether there was a potential connection between exotic animal

content and perception. We have further studies planned where data will be collected based on

hypotheses.

2.1 Thank you for your suggestion. The reviewer raises a valid point. In future, if we were to perform

further statistical analysis, we would take this into account. However, at this stage the study was

designed as a descriptive observational study. The aim was to analysis the relationship between

sentiment (as an indication of public perception) and species conservation status and explore the

association with the exotic pet trade. We wanted to avoid selection bias within the study. Being a

descriptive study, we wanted to observe this potential relationship to suggest that it may be an area

which could warrant further investigation. At this stage, the addition of the suggested model is

outside the scope of this paper.

2.2 We believe this comparison would be interesting, however the aim of the study was to evaluate

exotic animals in ‘free handling’ situations, wherein they featured some sort of interaction with

humans. Therefore, the way we view it, the interaction with humans was more important than the

setting, where we were aiming to imply that the setting was not necessarily as important in how

these animals were viewed in this context (i.e. animals seen in captivity could still be considered

desirable and promote demand for these animals as pets). We agree that this was not clearly

explained within the discussion and therefore have modified it to account for this (line 311-315).

3. Thank you for highlighting the ambiguity associated with this point. We have added some

additional description to explain why emoji analysis was important within this research (see line 288-

296). Firstly, we argue that it can be difficult to gauge emotion from words alone, where emojis act

as metadata. Additionally, emojis represent a global language, with the aim of avoiding cultural bias

and misinterpretations of words detected in non-English languages.

Introduction

1. We agree with the reviewer to make the link between social media and the exotic pet trade

stronger. We have revised the flow of the introduction, whilst ensuring the main points (as pointed

out by the reviewer) have been explicitly addressed.

2. We agree with the suggestion to make the link between YouTube and the popularity of animal

videos and have added a comment to address this.

3. In accordance with your comment, we have included further information as to why sentiment

analysis techniques were chosen and why it was relevant to the study (see lines 64-71).

Line 32: The text has been edited as per the suggestion.

Line 32-34: These changes have been applied.

Line 32-39: A statement has been added to summarise this paragraph.

Line 40-41: We have made changes to the flow of the paragraph, as requested.

Line 42: This line has been edited to avoid confusion.

Line 40-47: A closing point has been added to this paragraph to help strengthen the link between the

exotic pet trade and social media.

Line 51: We have changed ‘insufficiently regulated’ to ‘unregulated’ to avoid confusion.

Line 48-60: A new topic sentence which better addresses the main points of the paragraph has been

included.

Line 61-64: We have altered the layout of the aim accordingly.

Line 64-66: Upon review, we agree that this statement was misplaced. We have moved it into a

more appropriate position within the aim (line 35-36).

Methods

1. Further information regarding why the specific lexicons were chosen has now been included

within the methodology (see line 131-140 and line 155-158). After exploring various dictionaries, we

found that these lexicons provided the most appropriate information for our study.

2. Analysis was predominately based on evaluating sentiment trends generated and depicted

through gglot2. In accordance with this, we firstly noted no statistical difference between the IUCN

Red List Classifications in regards to sentiment (as shown in figure 2a and 2b) which suggested no

clear distinction between the categories (interpreted as lack of public awareness of the significance

of conservation status). In regards to changes in sentiment trends over time, we observed that

sentiment remained consistently positive throughout time and was significantly different from zero

based on the application of 95% confidence intervals (except for primate text sentiment in 2015,

figure 3b).

Line 90: This has now been rewritten.

Line 91: You are correct in that tuber::get_comments( ) typically only extracts 100 comments at a

time, however we designed code to bypass this, enabling the function to effectively be repeated to

extract all comments. A comment has been included within the paper to address this (see line 107-

109).

Line 109-111: Clarification regarding the sentiment parameters and lexicon selection has been

included.

Line 120 – 121: The emoji sentiment lexicon published by Novak et al. was chosen as it provided

suitable code to complement the requirements of the study and was novel work not detailed 

elsewhere when the analysis was being carried out. The code itself was not altered but was merely

written into R. A statement has been included in the methodology to describe this.

Results

1. As stated previously, although it would have been interesting to analyse the different response to

captive animals compared to pets, the most important point is that regardless of the setting the

animals were depicted in the interaction with humans (‘free handling’). The idea behind this was

that we wanted to suggest that despite the living situation of the animal (whether it was in a zoo or

as a pet), public perception was similar and both contributed to increased demand for these animals.

This was perhaps not very clear; hence, some revisions have been made to make this point stronger.

2. A table has been added to the supporting information regarding the number of comments

extracted. If a user made multiple comments within the comments section, these were all included

(as all comments were extracted). This includes both primary comments (main comments made) and

secondary comments (comments made underneath primary comments). If the creator commented

within the comments section, these were also included. A description has been included within the

methods (line 109-111).

Table 1: The table legend has been updated to include further information, as requested.

Line 155-160: This information has been moved to the methods section.

Line 192 – 194: Thank you for your suggestion. However, upon analysing the boxplots, the results

were not statistically significant from zero (with all plots intersecting zero), therefore the standard

deviation was not included. The analysis was predominately based on the positive distribution of

data. One of the main findings from the box plots was also surrounding how the classification groups

overlap, suggesting the public either did not recognise the conservation classification associated

with the species shown or did not care. 95% confidence intervals were applied and have been

included within the supplementary code file.

Line 186: Thank you for highlighting this uncertainty. The methods section has been updated to

make the sampling method clearer. A simple random sample was taken from each wild cat species

using the set.seen(n) function in R to produce the 10,000 comments. Sub-sampling involved ensuring

these comments were refined to only those containing at least one emoji (n>0) to provide a more

meaningful analysis.

Line 186-189: This section has been revised.

Line 192-193: This section has also been revised.

Discussion

1. The additional headers within the discussion have been removed to help improve flow, as

recommended. Further editing has been undertaken to try and make the aims clearer to the reader.

2. Thank you for highlighting this. We have aimed to re-emphasise the data and findings within the

discussion to address this.

Line 225: As now stated within the discussion, ‘YouTube® comments serve as a gauge of public

perception due to its wide-reaching accessibility and popularity, where several studies have

demonstrated the impact social media platforms have on public opinion’ (line 271-273). Therefore,

we believe that for the purposes of this study the comments extracted represented public opinion

surrounding the depiction of exotic animals in YouTube videos.

Line 225-230: The beginning of the discussion has been revised to ensure better flow.

Line 233-234: Some of this information has been moved to the methods section as requested,

whereas the interpretation remains within the discussion.

233-241: Thank you for your comment. We have aimed to incorporate the limitations of the

sentiment software with the aims of our study, as suggested. In regards to the identification of

sarcasm, it was difficult to derive a subset of comments which demonstrated this given the nature of

our data, however studies have been referenced regarding which emojis are commonly used in the

expression of sarcasm as a generalised example (line 293-295).

Line 304-331: Thank you for your insight regarding the recommendations made, we were unaware

of projects such as Wildbrook and after some research have decided to include it within the

discussion. Additional simpler changes have also been suggested, however there are likely other

approaches which could be considered also.

Figures

Changes have been made to the figures, as suggested.

---

## [Decision Letter · Decision Letter 1]

30 Dec 2020

PONE-D-20-17651R1

Is YouTube® promoting the exotic pet trade?

Analysis of the global public perception of popular YouTube® videos featuring threatened exotic animals.

PLOS ONE

Dear Dr. Chaber,

Thank you for submitting your manuscript to PLOS ONE. After careful consideration, we feel that it has merit but does not fully meet PLOS ONE’s publication criteria as it currently stands. Therefore, we invite you to submit a revised version of the manuscript that addresses the points raised during the review process.

Dear Dr. Chaber,

I would like to apologise for the delay in rendering a decision on the revised version of your submission PONE-D-20-17651R1-Is YouTube® promoting the exotic pet trade? Analysis of the global public perception of popular YouTube® videos featuring threatened exotic animals. Reaching an opinion on your manuscript has been challenging, since the two reviewers who assessed the revised version radically disagreed on their decision, one of them even recommending rejection. Consequently, I had then to search for a third opinion. As you will see below, major concerns remain preventing the publication of your manuscript. However, the reviewers and myself still feel than your research has potential to become an article worth being published in PLoS ONE. I therefore recommend you to undertake the revisions suggested by the reviewers and thoroughly revise your work to not only address the issues the reviewers have raised, but to substantially modify the study beyond the descriptive analysis, as suggested by the reviewers, including references and following examples such as Fidino et al. 2018 (Fidino, M., Herr, S. W., & Magle, S. B. 2018. Assessing online opinions of wildlife through social media. Human Dimensions of Wildlife, 23(5), 482-490).

If failing to produce such a relevant modification in your manuscript, the most likely outcome of a  second revised version would be rejection. I consequently encourage you to squeeze your database and take the most out of it through roust meaningful analyses following the reviewers' advice in order to gain final acceptance for publication in PLoS ONE.

Best luck when carrying out your review.

We look forward to receiving your revised manuscript.

Kind regards,

Jorge Ramón López-Olvera

Academic Editor

PLOS ONE

Additional Editor Comments (if provided):

Dear Dr. Chaber,

I would like to apologise for the delay in rendering a decision on the revised version of your submission PONE-D-20-17651R1-Is YouTube® promoting the exotic pet trade? Analysis of the global public perception of popular YouTube® videos featuring threatened exotic animals. Reaching an opinion on your manuscript has been challenging, since the two reviewers who assessed the revised version radically disagreed on their decision, one of them even recommending rejection. Consequently, I had then to search for a third opinion. As you will see below, major concerns remain preventing the publication of your manuscript. However, the reviewers and myself still feel than your research has potential to become an article worth being published in PLoS ONE. I therefore recommend you to undertake the revisions suggested by the reviewers and thoroughly revise your work to not only address the issues the reviewers have raised, but to substantially modify the study beyond the descriptive analysis, as suggested by the reviewers, including references and following examples such as Fidino et al. 2018 (Fidino, M., Herr, S. W., & Magle, S. B. 2018. Assessing online opinions of wildlife through social media. Human Dimensions of Wildlife, 23(5), 482-490).

If failing to produce such a relevant modification in your manuscript, the most likely outcome of a second revised version would be rejection. I consequently encourage you to squeeze your database and take the most out of it through roust meaningful analyses following the reviewers' advice in order to gain final acceptance for publication in PLoS ONE.

Best luck when carrying out your review.

Reviewers' comments:

Reviewer's Responses to Questions

**Comments to the Author**

1. If the authors have adequately addressed your comments raised in a previous round of review and you feel that this manuscript is now acceptable for publication, you may indicate that here to bypass the “Comments to the Author” section, enter your conflict of interest statement in the “Confidential to Editor” section, and submit your "Accept" recommendation.

Reviewer #2: All comments have been addressed

Reviewer #3: (No Response)

Reviewer #4: (No Response)

2. Is the manuscript technically sound, and do the data support the conclusions?

Reviewer #2: Yes

Reviewer #3: No

Reviewer #4: Partly

3. Has the statistical analysis been performed appropriately and rigorously? 

Reviewer #2: Yes

Reviewer #3: No

Reviewer #4: No

4. Have the authors made all data underlying the findings in their manuscript fully available?

Reviewer #2: Yes

Reviewer #3: (No Response)

Reviewer #4: Yes

5. Is the manuscript presented in an intelligible fashion and written in standard English?

Reviewer #2: Yes

Reviewer #3: Yes

Reviewer #4: Yes

6. Review Comments to the Author

Reviewer #2: You have addressed all my comments. I commend you on a very interesting paper and look forward to more - Jill R D MacKay

Reviewer #3: There were two main issues that I brought up with my previous review that the authors did not address.

First, the authors contend that perceptions and opinions are the same. This is wrong. Perceptions are what someone sees, how they interpret, what they smell, etc. Opinions are a view or judgement about a topic, and may not necessarily be based on knowledge of a topic. These words are not interchangeable, they mean different things in the literature and the authors have not convinced me, in this case, that the words are interchangeable.

Second, the authors state that they did not do any statistical analysis because they had no hypotheses to test given that this was a descriptive observational studies. This is an inadequate response. Observational studies, even if they do not have a set hypothesis (or hypotheses) very often require statistics. I believe that these data will require some statistical analysis, especially to control for the large-scale variability in the number of comments between videos. The current analysis, in this case, certainly functions as a "first step" in what the authors should have done for this paper. They plotted out some of the data. However, that is not the end point of this analysis. The unequal sample sizes among videos, combined with the fact that topics of the videos are different, no doubt has a strong influence on a mean response. Nevertheless, the authors have combined all the the data for a number of analyses (e.g., mean sentiment response), which essentially assumes that all content of videos is equivalent.

Even without set hypotheses, inferential statistics can still be extremely useful because they can provide a comparison of effect size among categories or features of a dataset that are more correlated to a given response variable. This would be incredibly helpful for future hypothesis generation. Using multi-model inference, for example, would be a great way to select for variables that explain more variability in the data than others (i.e., provide feature selection, make comparisons, etc.). As an example, even being able to statistically compare the number of comments made among videos of different species (line 185) would be better than simply making a statement about the given sample. In it's current form, I do not know if the results are essentially being overweighed by one or two videos or if their sample is representative of the overall population of videos online.

Reviewer #4: I think the topic of investigation is of broad interest and it would be important to publish these results. That said, more work is necessary, especially in the presentation of the results. Some of the issue raised by previous reviewers are still not fully addressed (statistical analysis, hypotheses), and I provide more suggestions.

- Lines 72-76. I understand here the critique of the reviewers. Even if you want to have a descriptive paper, you inevitably test hypotheses since you expect a difference in perception of exotic pets based on their IUCN status, and you also expect a trend with time. You can justify why in here.

- Lines 189-190. Table S2 does not really show a right-skewed distribution. There is no index of skewness reported. The median value is much lower than the mean, so the majority of videos have lower comments than the mean (so probably left-skewed), and there seems to be outliers on the right. But I would expect a histogram or a skewness index if you talk about skewness.

- Table 1. For some genera, you probably have more than one species (it is not clear if you did not identify them). It should be clear how you handled with this frequent situation as for some genera (e.g. Macaca) you can have IUCN classifications ranging from Least Concern to Critically Endangered. So having Macaca spp. and the relative IUCN Red List category Endangered is probably incorrect unless you considered only Endangered species. That is true for other genera as well as you do not present the species. In the table, there is also a typo (Varecia variegata not variegate). If you have the genus you should also add spp. if you consider more species of the same genus (and of course they should all be with the same IUCN status).

- Lines 210-211. I do not understand this result. You selected the videos with wild cats and primates, so not sure what are these results relative to dogs. Do you mean that wild cats and primates had interactions with dogs in 17.9% of the videos? Can we know more about which species?

- Line 216. Likewise is not appropriate here as IUCN Red List and CITES are not going on the same direction.

- Text sentiment analysis section. I would expect a word frequency analysis, that can be interesting and quick to add, and can show more about the content of the comments.

- Lines 229-231. This statement would require a statistical analysis. You also need to consider the other issue relative to IUCN Red List status, comment to table 1.

- Lines 246-247. Here you need statistical analyses to confirm your statements.

- Line 249. It seems neutral as it is close to 0 (unless it is statistically significant from 0).

- Line 270-271. Here you report the same phrase as in the results but you do not explain why it is important, you do not discuss this result, you just repeat it.

- Lines 279-280. I do not see this from your results.

- Line 299. You do not really discuss the implications of your results. You found that the perceptions were positive, but what does it mean? What does it entail? You should move here the paragraph that is now almost at the end of the discussion

- Line 301. You did not really test this

- Line 306. Did you recognize the species? As there are so many macaques with so many different statuses.

- Lines 359-374. I think this paragraph should go above as this is your main discussion on your findings.

- Line 369. But this is only relative to slow lorises, how can this impact your analysis as you have many species and only 7 videos of pygmy slow lorises.

- Line 420. IUCN Red List status ranged from Least Concern to Critically Endangered, so not all the species you considered are threatened. You should rephrase it.

- Figure 2a. The IUCN categories should be upper case. In the x axis label you say Mean Sentiment Score but you show box plots (i.e. median, quartiles, range, and outliers). I suggest changing the figures to have a better presentation, maybe use theme(bw), can try different themes in ggplot2.

7. PLOS authors have the option to publish the peer review history of their article (what does this mean?). If published, this will include your full peer review and any attached files.

Reviewer #2: **Yes: **Jill R D MacKay

Reviewer #3: No

Reviewer #4: No

---

## [Author Response · Author response to Decision Letter 1]

18 Feb 2021

Dear Reviewer,

Thank you very much for your valuable comments, we have substantially modified the manuscript following your suggestions and hope this improved version will meet your expectation.

Best regards,

ALC

The reviewer’ comments are addressed below the initial comments.

Reviewer #4: I think the topic of investigation is of broad interest and it would be important to publish these results. That said, more work is necessary, especially in the presentation of the results. Some of the issue raised by previous reviewers are still not fully addressed (statistical analysis, hypotheses), and I provide more suggestions.

=>In order to address these comments we have included mixed models to assess which variables were associated with text and emoji sentiments

- Lines 72-76. I understand here the critique of the reviewers. Even if you want to have a descriptive paper, you inevitably test hypotheses since you expect a difference in perception of exotic pets based on their IUCN status, and you also expect a trend with time. You can justify why in here.

=>We have included more analyses in the paper and have reworded objectives accordingly.

- Lines 189-190. Table S2 does not really show a right-skewed distribution. There is no index of skewness reported. The median value is much lower than the mean, so the majority of videos have lower comments than the mean (so probably left-skewed), and there seems to be outliers on the right. But I would expect a histogram or a skewness index if you talk about skewness.

=>This is correct, we have hence reworded this

- Table 1. For some genera, you probably have more than one species (it is not clear if you did not identify them). It should be clear how you handled with this frequent situation as for some genera (e.g. Macaca) you can have IUCN classifications ranging from Least Concern to Critically Endangered. So having Macaca spp. and the relative IUCN Red List category Endangered is probably incorrect unless you considered only Endangered species. That is true for other genera as well as you do not present the species. In the table, there is also a typo (Varecia variegata not variegate). If you have the genus you should also add spp. if you consider more species of the same genus (and of course they should all be with the same IUCN status).

=>We have clarified the IUCN classifications of this group and included this in the paper (and corrected the spelling mistake; thank you for noticing).But, you are correct, there were several Macaca species, for this situation, IUCN listing was based on most severe status of identified species

- Lines 210-211. I do not understand this result. You selected the videos with wild cats and primates, so not sure what are these results relative to dogs. Do you mean that wild cats and primates had interactions with dogs in 17.9% of the videos? Can we know more about which species?

=>The interpretation is correct, this has now been reworded – line 220 now

- Line 216. Likewise is not appropriate here as IUCN Red List and CITES are not going on the same direction.

=>This has been reworded to correct (line 225)

- Text sentiment analysis section. I would expect a word frequency analysis, that can be interesting and quick to add, and can show more about the content of the comments.

=>To address this we have included the mixed models for text and emoji sentiment

- Lines 229-231. This statement would require a statistical analysis. You also need to consider the other issue relative to IUCN Red List status, comment to table 1.

=>We have reworded this and changed the figure – now starting line 245

- Lines 246-247. Here you need statistical analyses to confirm your statements.

=>This has been reworded (line 276 start)

- Line 249. It seems neutral as it is close to 0 (unless it is statistically significant from 0).

=>This has been deleted and results presented based on the mixed models

- Line 270-271. Here you report the same phrase as in the results but you do not explain why it is important, you do not discuss this result, you just repeat it.

=>Added interpretation - Now line 305

- Lines 279-280. I do not see this from your results.

=>These results are included in line 230

- Line 299. You do not really discuss the implications of your results. You found that the perceptions were positive, but what does it mean? What does it entail? You should move here the paragraph that is now almost at the end of the discussion

=>We have moved the paragraph as suggested (line 345) and combined with discussion as per mixed method models

- Line 301. You did not really test this

=>This is true, we have now deleted this statement

- Line 306. Did you recognize the species? As there are so many macaques with so many different statuses.

=>The results are based on the identified species (as per mentioned above)

- Lines 359-374. I think this paragraph should go above as this is your main discussion on your findings.

=>Moved as suggested - line 344

- Line 369. But this is only relative to slow lorises, how can this impact your analysis as you have many species and only 7 videos of pygmy slow lorises.

=>This was probably not clear from the results included. We suggested this due to the results in the new and more detailed figures included now. This includes figure 2 (slow loris the only species with negative sentiment) and figure 5 (reduction in primate sentiment in 2015). 

- Line 420. IUCN Red List status ranged from Least Concern to Critically Endangered, so not all the species you considered are threatened. You should rephrase it.

=>This comment is correct; however, the results show that all groups on the IUCN red list have a positive median sentiment. We have tried to clarify this. 

- Figure 2a. The IUCN categories should be upper case. In the x axis label you say Mean Sentiment Score but you show box plots (i.e. median, quartiles, range, and outliers). I suggest changing the figures to have a better presentation, maybe use theme(bw), can try different themes in ggplot2.

=>Figure has been changed (now fig 3)

---

## [Decision Letter · Decision Letter 2]

8 Mar 2021

PONE-D-20-17651R2

Is YouTube® promoting the exotic pet trade?

Analysis of the global public perception of popular YouTube® videos featuring threatened exotic animals.

PLOS ONE

Dear Dr. Chaber,

Thank you for submitting your manuscript to PLOS ONE. After careful consideration, we feel that it has merit but does not fully meet PLOS ONE’s publication criteria as it currently stands. Therefore, we invite you to submit a revised version of the manuscript that addresses the points raised during the review process.

I have now received the comments by one of the reviewers who assessed the first revision of your manuscript, and while he founds that the manuscript has improved considerably, he still points out some minor issues to address before considering your submission fully publishable in PLoS ONE.

I suggest you to undertake the suggested modifications in order to gain final acceptance for publications. Best regards,

We look forward to receiving your revised manuscript.

Kind regards,

Jorge Ramón López-Olvera

Academic Editor

PLOS ONE

Journal Requirements:

Additional Editor Comments (if provided):

Dear Dr. Chaber,

I have now received the comments by one of the reviewers who assessed the first revision of your manuscript, and while he founds that the manuscript has improved considerably, he still points out some minor issues to address before considering your submission fully publishable in PLoS ONE.

I suggest you to undertake the suggested modifications in order to gain final acceptance for publications. Best regards,

Reviewers' comments:

Reviewer's Responses to Questions

**Comments to the Author**

1. If the authors have adequately addressed your comments raised in a previous round of review and you feel that this manuscript is now acceptable for publication, you may indicate that here to bypass the “Comments to the Author” section, enter your conflict of interest statement in the “Confidential to Editor” section, and submit your "Accept" recommendation.

Reviewer #4: All comments have been addressed

2. Is the manuscript technically sound, and do the data support the conclusions?

Reviewer #4: Yes

3. Has the statistical analysis been performed appropriately and rigorously? 

Reviewer #4: Yes

4. Have the authors made all data underlying the findings in their manuscript fully available?

Reviewer #4: Yes

5. Is the manuscript presented in an intelligible fashion and written in standard English?

Reviewer #4: Yes

6. Review Comments to the Author

Reviewer #4: The authors did a good job in addressing the comments. In my opinion, the paper is now almost ready to be published. I just have a few more suggestions:

- In the abstract you need to integrate your new findings.

- Line 113. Here you have an open bracket, need to edit the sentence somehow.

- Line 116. Data is the plural of datum, please check throughout the manuscript that the verbs are correct.

- Lines 175-179. Add the command used to run the mixed model. Also, it is not clear to me what you used as fit function for your response variables. Did you use general or generalised linear mixed models? Also, you do not add a random effect to deal with the repeated measurement. You add a random effect to deal with the fact that your data are taken from different videos. The concept of repeated measurement and the concept of random effect are not the same.

- Line 190. add spp. after Macaca.

- In the results, your modelling sections are now disconnected from the other sections of the results. It is a bit confusing. I suggest to integrate them better, you do not need separate sections.

- Line 253. It should be 0.6. Also, it should be 2011 as 2010 is not significantly different than 2006.

- Line 312. Missing s in slow loris.

- Line 321. It is significant so might want to rephrase with something "stronger" than appeared to have.

- Lines 343-347. This paragraph needs some link to literature or to be integrated into another paragraph. It is now too short and unreferenced to be a separate paragraph.

- Lines 324, 344. It is weird to discuss "data not shown". It would be better to add them as supplementary materials.

- Lines 430-432. It seems that Least Concern animals have more positive texts associated to the videos than the other IUCN categories? Please check again that your discussion and conclusion reflects your new results.

- Figure legends. Quantiles should be quartiles in the figure legends (and check the rest of the MS)

7. PLOS authors have the option to publish the peer review history of their article (what does this mean?). If published, this will include your full peer review and any attached files.

Reviewer #4: No

---

## [Author Response · Author response to Decision Letter 2]

25 Mar 2021

Response to Reviewer 4

Abstract: The abstract has been altered to incorporate the new findings.

Line 113: This was corrected. 

Line 116. The terminology was checked accordingly and we believe it to be correct now. 

Line 175-179: The command used to run the mixed model has now been added to the methods section (line 181-183). We used the lmer function which is a general linear mixed model. Furthermore, we understand that repeat measures and random intercepts are not the same, but we found no evidence for the need of a temporal effect. 

Line 190: Thank you for pointing this out, it has been corrected.

Results: Thank you for your suggestion, the individual sections have been reduced to integrate the results better. 

Line 253. Thankyou for noticing this, it has been corrected.

Line 312: This has also been corrected.

Line 321. This has been rephrased with more certainty.

Lines 343-347. This paragraph has now been integrated into the results section where it appears to be more suitable. 

Lines 324, 344: Data can be obtained from the authors as required upon request, hence was not included within the supplementary information. 

Lines 430-432: The higher median text sentiment in ‘Least Concern’ species was mentioned earlier in the text (line 333-335), wherein the conclusion has now been reworded to make this clearer to the reader.

Figure legends: Thank you for pointing this out, ‘quantiles’ has been changed to ‘quartiles’ throughout the manuscript. 

Thank you for your time and suggestions to help improve our manuscript.

---

## [Decision Letter · Decision Letter 3]

29 Mar 2021

Is YouTube® promoting the exotic pet trade?

Analysis of the global public perception of popular YouTube® videos featuring threatened exotic animals.

PONE-D-20-17651R3

Dear Dr. Chaber,

We’re pleased to inform you that your manuscript has been judged scientifically suitable for publication and will be formally accepted for publication once it meets all outstanding technical requirements.

Kind regards,

Jorge Ramón López-Olvera

Academic Editor

PLOS ONE

Additional Editor Comments (optional):

Dear Dr. Chaber,

I have now received the comments on your last submission from the reviewer who assessed the last version and i am pleased to inform you that your manuscript can be now considered suitable for publication in PLoS ONE.

Thank you very much for considering our journal to publish your research and congratulations for the work done and the results obtained.

Best regards,

Reviewers' comments:

Reviewer's Responses to Questions

**Comments to the Author**

1. If the authors have adequately addressed your comments raised in a previous round of review and you feel that this manuscript is now acceptable for publication, you may indicate that here to bypass the “Comments to the Author” section, enter your conflict of interest statement in the “Confidential to Editor” section, and submit your "Accept" recommendation.

Reviewer #4: All comments have been addressed

2. Is the manuscript technically sound, and do the data support the conclusions?

Reviewer #4: Yes

3. Has the statistical analysis been performed appropriately and rigorously? 

Reviewer #4: Yes

4. Have the authors made all data underlying the findings in their manuscript fully available?

Reviewer #4: Yes

5. Is the manuscript presented in an intelligible fashion and written in standard English?

Reviewer #4: Yes

6. Review Comments to the Author

Reviewer #4: I had a quick check at the sections the authors modified and I am happy with the edits. I think the paper is interesting and it is now coherent and ready to be published.

7. PLOS authors have the option to publish the peer review history of their article (what does this mean?). If published, this will include your full peer review and any attached files.

Reviewer #4: No

---

## [Editor Report · Acceptance letter]

1 Apr 2021

PONE-D-20-17651R3 

Is YouTube promoting the exotic pet trade? Analysis of the global public perception of popular YouTube videos featuring threatened exotic animals. 

Dear Dr. Chaber:

I'm pleased to inform you that your manuscript has been deemed suitable for publication in PLOS ONE. Congratulations! Your manuscript is now with our production department. 

Kind regards, 

on behalf of

Dr. Jorge Ramón López-Olvera 

Academic Editor

PLOS ONE